# A liquid metal-based module emulating the intelligent preying logic of flytrap

Yuanyuan Yang [1] & Yajing Shen [2,3] ✉

Plant species like the Venus flytrap possess unique abilities to intelligently respond to various external stimuli, ensuring successful prey capture. Their nerve-devoided structure provides valuable insights for exploring natural intelligence and constructing intelligent systems solely from materials, but limited knowledge is currently available and the engineering realization of such concept remains a significant challenge. Drawing upon the flytrap's action potential resulting from ion diffusion, we propose a signal accumulation/attenuation model and a corresponding liquid metal-based logic module, which operates on the basis of the shape change of liquid metal within a sodium hydroxide buffer solution. The module itself exhibits memory and counting properties without involving any other electronic components, intelligently responding to various stimulus sequences, and reproducing the flytrap's most logical function. We also demonstrate and forecast its potential as a moving window integration-based high-pass filter, artificial synapse in neural networks, and other related applications. This research provides a fresh perspective on comprehending the intelligence inherent in nature and its realization through physical structures, which is expected to inspire logic device development in a broad engineering field.

The Venus flytrap, a carnivorous plant, has evolved an intelligent sensing system for capturing prey, which can differentiate various external stimuli such as fast single touch, long single touch, fast consecutive double touches, and fast discrete double touches[1]. Intriguingly, scientists found that its sensory hair-based sensing system possesses features akin to memory and counting, i.e., the Venus flytrap can count the number of stimuli on sensory hairs and remember them for a short-term duration, commanding the plant's trap to close if the insect taps twice within approximately half a minute[2] (Fig. 1a). It is perplexing that flytrap's intelligence arises purely from the proper organization of simple materials and structure, in stark contrast to the complex nervous systems of animals. Investigating and emulating the plant's behavior could provide new insights into understanding the intelligence of nature, which is significant for both fundamental scientific research and practical engineering applications.

To mimic the flytrap's snapping motion, scientists have proposed several artificial structures and realized the autonomous opening and closure of artificial Venus flytraps responding to different external stimuli, such as electric, light, and thermal field[3–8]. Some researchers also proposed a bistable bending actuators-based flytrap, which could implement snapping motion through the quick transfer between the two stable shapes[9]. However, the control and response mode in these works is quite straightforward, i.e., the system receives a single stimulus and then close, which is far from the intelligent behavior of real Venus flytraps. Recently, some electrical modulating units have been developed to connect with real flytraps, paving a new way for the integration of physical systems with plants[10–12]. Yet, these researches were focusing on the bio-integration interface rather than a flytrap-like intelligent system. Despite these achievements in artificial flytraps, it remains challenging to replicate the complex interactions between the

[1]School of Aerospace Engineering, Xiamen University, Xiamen, China. [2]Department of Electronic and Computer Engineering, The Hong Kong University of Science and Technology, Clear Water Bay, Hong Kong, China. [3]Center for Smart Manufacturing, The Hong Kong University of Science and Technology, Clear Water Bay, Hong Kong, China. ✉e-mail: eeyajing@ust.hk

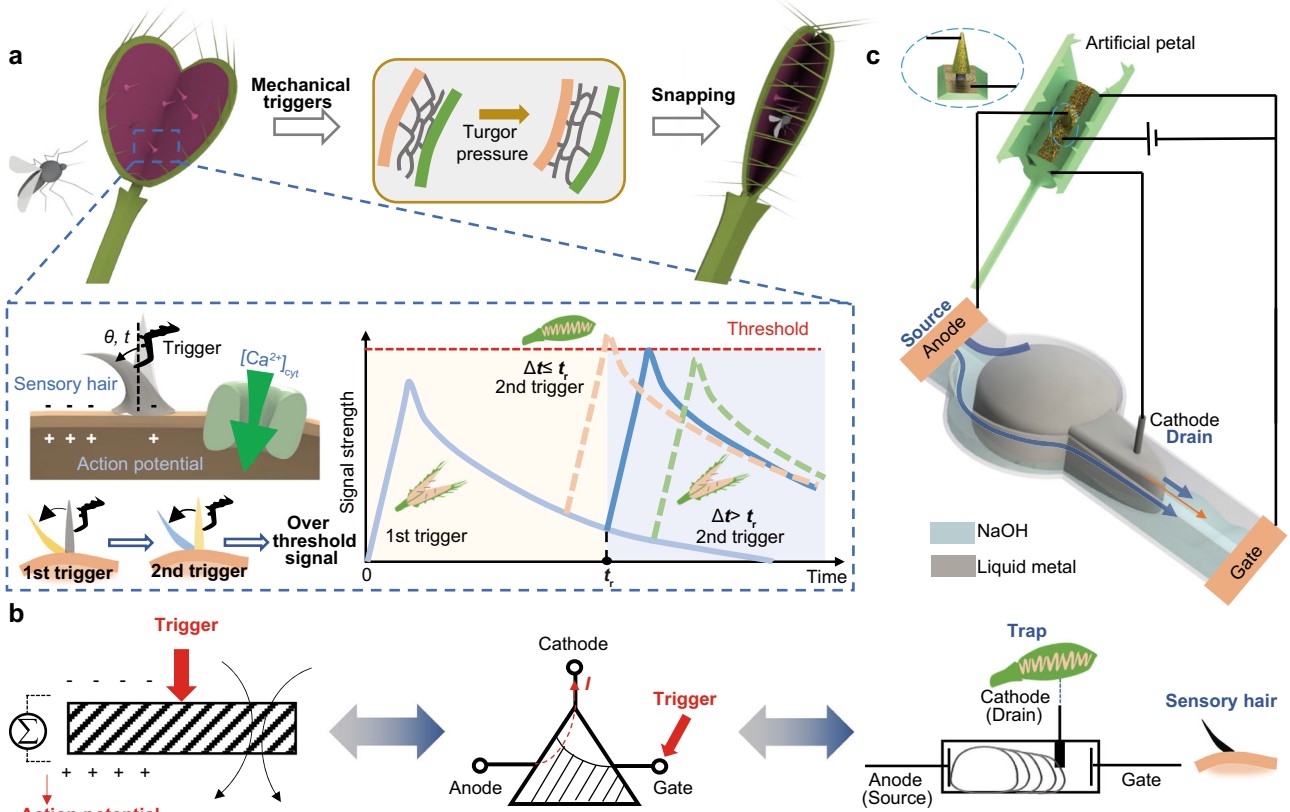

**Fig. 1 | Venus flytrap-inspired liquid metal-based logic module. a** The Venus flytrap could generate an electric signal after receiving mechanical stimuli to sensory hair, leading to the increase of cytoplasmic calcium ($[Ca^{2+}]_{cyt}$) concentration. The strength of the electric signal gradually decreased after the first stimulus, then a second stimulus is required to increase the signal to a higher level, meeting the threshold that is correlated to the leaf blade closure. Once the time interval between two stimuli $\Delta t$ exceeds the threshold $t_r$, the strength increase induced by the second stimulus will be insufficient to meet the putative threshold for leaf closure. **b** The ion concentration could be abstracted out as the signal accumulation/attenuation time-dependent ion diffusion (SAA) model. The SAA model suggests itself to be a three-node embedded system, the conductive medium of which is required with time-dependent positive and negative potential variation ability corresponding to the gate trigger. Inspired by that, a liquid metal-based logic module (LLM) is proposed. *I*: current output. **c** The liquid system is designed based on amorphous liquid metal, acting as the controller for the artificial flytrap. Similar to the ion diffusion mechanism within a flytrap, the liquid metal alters its shape based on its surface tension variation. Once the length of liquid metal reaches the threshold, the artificial flytrap petals would receive the corresponding electric signals to implement the closure process.

artificial flytraps and diverse environmental stimuli, not to mention the development of Venus flytrap-like intelligent components.

Liquid metal, as a kind of eutectic alloy at room temperature, exhibits great potentials as the materials to develop soft electronic components owing to its unique properties of desirable flexibility, high electrical conductivity, and low toxicity. Compared with the conventional conductive material, the liquid metal could be manipulated by external triggers (e.g., electrical, chemical, etc.), and then transform between different morphologies or move[13]. Due to these characters, several intriguing liquid metal-based electric devices have been proposed, such as liquid-metal based-electrical switches and logic components[14–16]. These functions are mainly based on the motions of liquid metal, i.e., coalesce, separation, and migration, which provide changeable states as electrical 0 and 1. Nevertheless, the two-state feature could not meet the requirement for emulating the Venus flytrap-liked logic since the bio-signals are changing dynamically. Thus, the time-related shape deformation of liquid metal should be taken into account. It could be regulated by external triggers, which is quite similar to the trigger-controlled variation of bio-signals.

In this study, we present a signal accumulation/attenuation (SAA) model derived from the Venus flytrap's behavior and introduce a liquid metal-based logic module (LLM) accordingly (Fig. 1b, c). The module comprises three nodes (anode, gate, cathode) with liquid metal in

NaOH solution as the conductive medium (Fig. 2a). The potential difference across the anode and cathode is determined by the trigger signal applied on the gate (positively correlated) and the capillary resistance of the system (negatively correlated), resulting in a time-depended characteristic. Our findings indicate that the LLM itself can memorize the duration and interval of stimuli, count the accumulated signal, and exhibit a remarkable logic function akin to the Venus flytrap. This work not only provides insights into the emulation of the intelligent behavior of plants but also demonstrates the potential of using bio signal-mimic design to develop autonomous systems for neuromorphic applications.

## Results
### Flytrap-inspired SAA model and LLM design
The Venus flytrap uses two intelligent strategies to distinguish the prey from other disturbances and generate enough electrical signals to close its trap, i.e., fast consecutive double touches and long single touch (Supplementary Movie 1). The behavior can be explained by the calcium dynamics-based signal memory effect. When a sensory hair is mechanically triggered, the concentration of cytoplasmic calcium ($[Ca^{2+}]_{cyt}$) within the membrane increases at the base of the hair. When two fast impulses occur in the sensory hair within the cooling down period (usually means the prey is a live bug worthy of consumption),

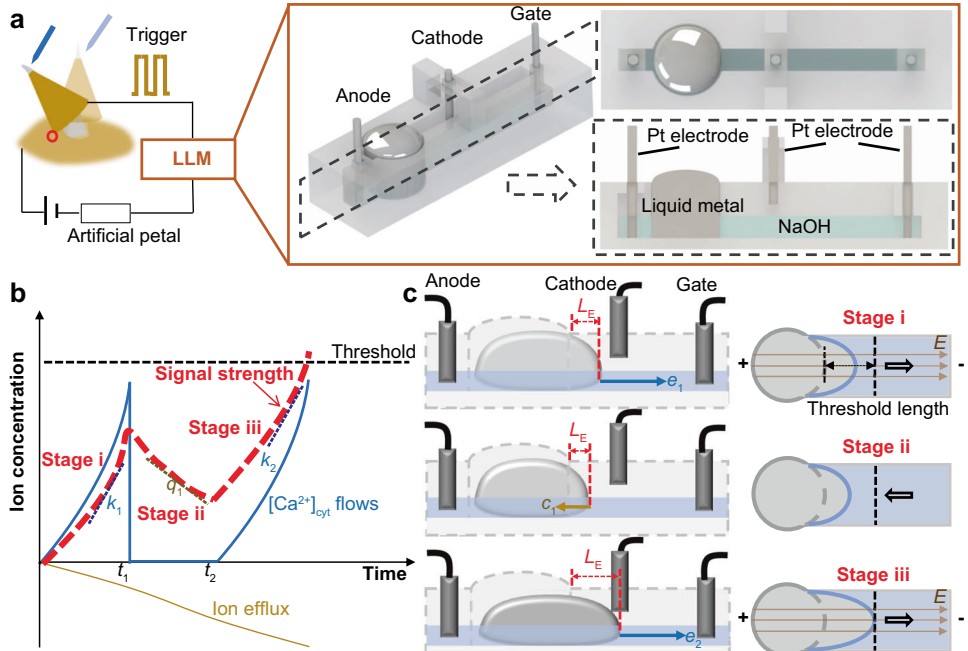

**Fig. 2 | Flytrap-inspired signal accumulation/attenuation (SAA) model and liquid metal-based logic module (LLM) design. a** The trigger-dependent closure of artificial petal could be implemented by the LLM. It mainly consists of a mushroom-shaped channel with a reservoir and a connected pathway. Liquid metal is placed in the reservoir and the rest of the cavity is filled with NaOH solution. Two Pt electrodes are inserted into the NaOH solution as the anode and gate, which are located on the left side and right side of the channel, respectively. Additionally, one Pt electrode between the anode and gate is designed as the cathode. **b** The ion concentration could be abstracted out as the SAA model. When a fast stimulus is applied on the sensory hair, the ion level increases (stage i, variation rate $k_1$) and then fades off at $t_1$ (stage ii, variation rate $q_1$), until a second stimulus is applied within the attenuation period at $t_2$ to accumulate it to exceed the threshold (stage iii, variation rate $k_2$). **c** The output of the liquid system depends on the shape deformation of the liquid metal filament. Two consecutive triggers could lead to the conducting of LLM that the length of liquid metal filament $L_E$ increases (stage i, variation rate $e_1$) and decreases (stage ii, variation rate $c_1$) when first electric potential ($E$) is applied and withdrawal, and then increases again as the second electric trigger applied (stage iii, variation rate $e_2$).

the $[Ca^{2+}]_{cyt}$ level increases and accumulates, triggering trap closure if it exceeds the threshold. Specifically, the first mechanical stimulus to a sensory hair increases $[Ca^{2+}]_{cyt}$, and the elevated $[Ca^{2+}]_{cyt}$ decreases after the first stimulus. The second stimulus additively increases $[Ca^{2+}]_{cyt}$ that meets a putative threshold for movement. This enables the flytrap to capture small animals that quickly move on its surface while avoiding incorrect stimuli such as wind and water drops. Additionally, the flytrap can respond to sustained loads, accumulating $[Ca^{2+}]_{cyt}$ to exceed the threshold and capture slow-moving animals like insect larvae and snails[1]. These fast consecutive double touches and long single touches make flytrap an effective predator.

Drawing inspiration from the calcium dynamics and closure mechanism of the flytrap, we abstract out a signal accumulation/attenuation (SAA) time-depended ion diffusion model. that

$$f(t) = \int_0^{t_1} k_1(t)dt - \int_{t_1}^{t_2} q_1(t)dt + \int_{t_2}^{t} k_2(t)dt, \quad (1)$$

where $f$ is the ion concentration difference from the initial balance state, $k_1$ and $k_2$ is the ion concentration variation rate caused by $[Ca^{2+}]_{cyt}$ level increase during the 1st and 2nd triggers, $q_1$ is ion concentration variation rate caused by the ion effluxion (i.e., potassium ion $K^+$), $t_1$ is the demarcation time between the accumulation and attenuation stages of 1st triggered-signal, $t_2$ is the time when 2nd trigger is applied. As shown in Fig. 2b, when a fast stimulus is applied on the sensory hair, the signal strength (red line, difference between ion flow in and ion efflux) increases (stage i) and then fades off (stage ii) until a second stimulus is applied within the attenuation period to accumulate it to exceed the threshold (stage iii). Conversely, if the sustained stimulus is large enough, the ion concentration can be accumulated to exceed the signal required for output. This suggests

that the system can define the duration and interval of stimuli based on accumulation and attenuation of ion signal, exhibiting distinct memory and counting ability.

Guided by the SAA model, we propose a liquid metal-based logic module (LLM) to mimic the intelligence of flytraps, i.e., a conductive medium-contained three-node module has time-depended positive and negative variation ability corresponding to the gate trigger. The LLM mainly consists of an open mushroom-shaped channel with a reservoir and a connected pathway. 100 µL Eutectic gallium–indium (EGaIn) is placed in the reservoir and the rest of the cavity is filled with 1 mol/L NaOH solution (1 mm in height). Two Pt electrodes are inserted into the NaOH solution as the anode (source of the LLM) and gate, which are located on the left side and right side of the channel, respectively. Liquid metal serves as the conductive medium to emulate the potential variation based on the electrochemically controlled capillary effect-induced length variation[17]. The stimulus signal from sensory hair is received by the anode and gate electrode in the form of potential difference. Additionally, one Pt electrode between the anode and gate is designed as the cathode (drain of the LLM) with a height of 1 mm above the NaOH liquid level, which is used to output signals to the trap.

The LLM relies on the interplay between the amorphous liquid metal filament and the flowing NaOH solution, where the positive and negative length variations of liquid metal filament $L_E$ are regulated by the electric trigger-depended surface tension variation, the capillary resistance, and the cohesion of liquid metal. This phenomenon can be modeled as a function of electric trigger and time, represented as

$$L_E(t) = \int_0^{t_1} e_1(t)dt - \int_{t_1}^{t_2} c_1(t)dt + \int_{t_2}^{t} e_2(t)dt, \quad (2)$$

where $e_1$ and $e_2$ are the length increase rates caused by the 1st and 2nd electric triggers-induced electrochemical effect, $c_1$ is the length decrease rate related to the cohesion of liquid metal and capillary resistance, $t_1$ is the time when the 1st gate trigger withdrew, $t_2$ is the time when the 2nd trigger is applied (Fig. 2c). As depicted, if $\int_0^{t_1} e_1(t) dt$ is large enough, indicating that long electric trigger is applied to the LLM, the liquid metal filament will extend enough to connect the cathode for signal output. Otherwise, two consecutive fast triggers are required that the length of the liquid metal filament increases (stage i) and decreases (stage ii) when the first trigger is applied and withdrawn, and then increases again as the second trigger is applied (stage iii).

## Liquid metal's time-depended shape deformation mechanism

To elucidate the characteristics of the LLM, we investigate the shape deformation mechanism of the liquid metal in the channel first. As a room-temperature eutectic alloy, the liquid metal exhibits amorphous, time-dependent, and stimulus-responsive properties. Upon immersion in NaOH solution, an electrical double layer (EDL) is formed at the interface between the solution and liquid metal. The EDL undergoes changes in its charge distribution in response to the electrical potential on the gate $V_G$, which generates an electric potential gradient along the liquid metal-electrolyte interface $V_E$. The relationship between the interface tension and the voltage difference across the EDL is represented by Lippmann's equation, that

$$\gamma = \gamma_0 - \frac{C}{2} V^2 \tag{3}$$

where $\gamma$ is the surface tension of the liquid metal, $\gamma_0$ is the intrinsic surface tension at the potential of zero charge, $C$ is the per-unit capacitance across the EDL, and $V$ is the electrical potential across liquid metal−electrolyte interface. The pressure difference across the EDL can be obtained from the Young−Laplace's equation:

$$P = \gamma \left( \frac{1}{r_1} + \frac{1}{r_2} \right) \tag{4}$$

where $P$ is the pressure difference between the electrolyte and the liquid metal, $r_1$ and $r_2$ are the radius of liquid metal. According to Eqs. (3) and (4), the surface tension should increase in the direction from anode to gate that the pressure on right part of liquid metal would be larger than left part, leading to an extending pressure $P_{LM}$ to the anode direction. Nevertheless, the existence of electrochemical effect would change the direction of surface tension gradient, especially for the liquid metal with large volumn[18,19]. Due to the voltage applied between anode and gate, the entrance of the pathway would attract a large amount of ions and promoting an oxidation reaction that leads to the formation of an oxide film on the liquid metal (Fig. 3a(i)). As the oxide film is hydrophilic, the surface tension would be highly reduced. The liquid metal would tend to increase its contact area with the surrounding electrolyte, resulting in the generation of extending pressure $P_{LM}$ to the gate direction that facilitates elongation of the liquid metal droplet within the channel. When $V_G < 5\,V$, surface tension-induced pressure $P_{LM}$ is not sufficient to overcome the capillary resistance $P_C$ (Fig. 3a(ii)), preventing the liquid metal from extending into the pathway. On the other hand, when $V_G \geq 5\,V$, $P_{LM}$ becomes strong enough to exceed $P_C$, enabling the liquid metal to gradually extend into the pathway, as shown in Fig. 3a(iii) and the red dashed line (form A) in Fig. 3b. The force analysis diagram of extending liquid metal is shown in Supplementary Fig. 1, which mainly include the electrochemical-induced surface tension force $F_{EC}$, the viscous force between the droplet and its surrounding electrolyte $F_V$, the friction between the liquid metal and the substrate $f_S$, and the friction between the liquid metal and the channel wall $f_W$. Therefore, during the extending process of liquid metal, the relevant factors corresponding to the $e_1$ and $e_2$ of

Eq. (2) could be specified as $F_{EC}, F_V, f_S$, and $f_W$. Since the electrochemical effect-induced surface tension reduction is positively correlated to $V_E$, the $F_{EC}$ as well as the extending speed of the liquid metal $v_l$ are positively related to $V_E$. Thus, a higher $V_G$ results in a higher $V_E$ and consequently higher $v_l$ (Supplementary Fig. 2). We also investigated the effect of the liquid metal's volume and pathway width $w$ on the extending speed of liquid metal $v_l$ (Supplementary Fig. 3, 4). This demonstrates that the LLM can be easily controlled by adjusting the device parameters in practical applications, providing adjustable options as an electrically stimulated acceptor for specific tasks on demand.

Under a continuous $V_G \geq 5\,V$, the LLM transitions to the ON state as the liquid metal extends and makes contact with the cathode wire, as indicated by the blue dashed line (form B) in Fig. 3b and the $L_E$ variation in Fig. 3c ($V_G = 7\,V$). The liquid metal is in contact with the anode at the first few seconds when voltage is applied to the gate and anode (Supplementary Fig. 5). Since the volume of liquid metal is constant, during the extending process of liquid metal filament, the left end of liquid metal would gradually separate with the anode. When the cathode is conducted, the liquid metal is not directly in contact with the anode while directly contacts the cathode. For the consideration that the liquid metal within the reservoir should be full enough to reach the cathode and still reserve a certain amount of it within the reservoir to drag the effluent liquid metal back after voltage off, the dimension of reservoir should be larger than the channel width. According to the experiments, the devices perform well over different reservoir dimensions (Supplementary Fig. 6) and in this work $r = 2.2\,mm$ is used for demonstration. At the cathode-contacted point, the forward motion of the liquid metal is halted by surface tension and physical blockage by the cathode wire. As a result, the liquid metal becomes trapped in the channel between the anode and cathode electrodes, effectively avoiding signal interference from the gate electrode. This stable configuration allows the LLM to function as a thyristor, controlling the conduction and nonconduction of the main channel in response to triggers from the gate.

To validate the physical model and evaluate the electric properties of LLM, a 1 kΩ resistor is connected to the main circuit, and a series of voltages ranging from 5 V to 9 V is applied to the gate. The I-V characteristic and conducting time $t_c$ of LLM, which is the duration from the initial state to the conducting state, are investigated. The results show that the current $Ic$ stabilizes quickly as the flow of the liquid metal becomes steady after achieving conduction, and its value is positively correlated with the gate voltage $V_G$ and the molarity of NaOH, in good agreement with the theoretical prediction (Fig. 3c, d, and Supplementary Fig. 7). Furthermore, the results demonstrate that the conducting time $t_c$ can be shortened by increasing the trigger voltage $V_G$ to accelerate the extending speed $v_l$ of liquid metal (Fig. 3e). For instance, by increasing $V_G$ from 5 V to 9 V, $t_c$ can be reduced by approximately 6 times, from 15.4 s to 2.5 s. This voltage-adjustable characteristic provides LLM with flexibility and controllability in practical applications as a designable time-delay electronic.

## The LLM's memory and counting properties for logic control

The time-depended shape deformation mechanism of liquid metal filament in the channel endorses the LLM memory property. To demonstrate this, we triggered the LLM within a short-time period ($\Delta t_{ON} = 3\,s$), which is shorter than the required conducting time $t_c$, using a gate voltage $V_G$ of 7 V. As shown in Fig. 4a(i) and (ii), when the gate voltage is turned off at t = 3 s, the extension of the liquid metal filament ceases immediately. However, intriguingly, the liquid metal filament remains in place for a certain period of time ($\Delta t_{ox} \approx 2.5\,s$), as shown in Fig. 4a(iii), rather than retracting immediately. This property enables the LLM to retain the memory of its extension length and continue elongating if a second triggering signal is received. This phenomenon is mainly governed by the adhesion of the oxide film to the channel, which will lead to a large friction force to channel wall $f_W$ and substrate $f_S$ (Supplementary Fig. 8). The relevant factors corresponding to the $c_1$ of Eq. (2) could be

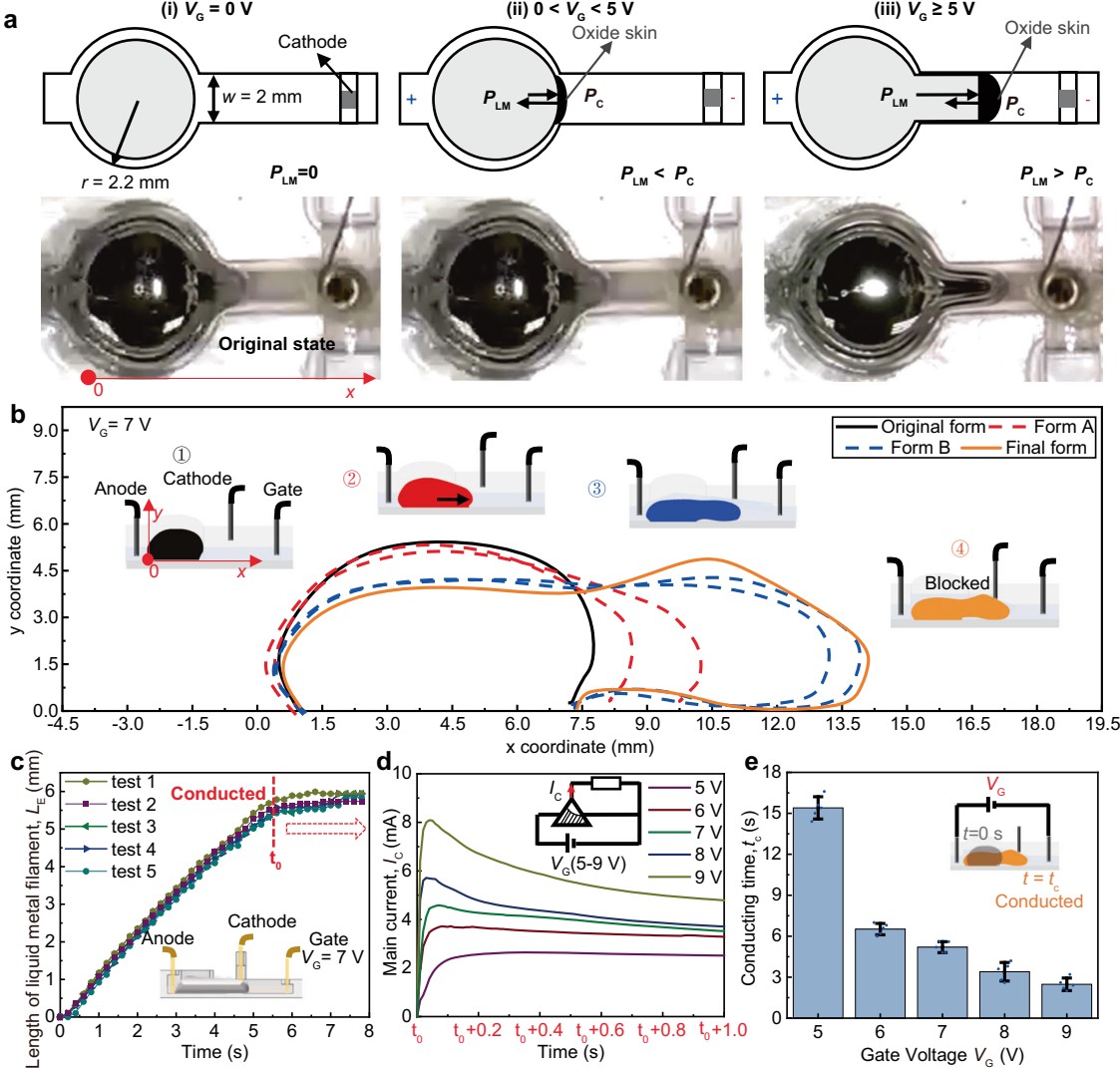

**Fig. 3 | Liquid metal's shape transformation-based working mechanism of LLM.**
**a** The shape deformation mechanism of liquid metal filament is based on the electrochemical effect. After applying gate voltage $V_G$, surface tension reduction occurs to actuate the shape deformation of the liquid metal filament, generating surface tension-induced pressure $P_{LM}$ to overcome the capillary resistance $P_C$. **b** The time-related shape deformation of liquid metal filament. **c** The length of the liquid metal filament would increase until touches the cathode at the time $t_0$ to conduct the device. **d** The output main current $I_C$ of conducted LLM is time-related and in positive relation with the gate voltage applied. **e** The conducting time $t_c$ required for different gate voltage inputs decreases with higher voltage applied. Data are presented as mean values ± standard deviation (SD), the number of independent experiments $n = 5$.

specified as $f_S$, $f_W$, viscous force from NaOH $F_V$, and the cohesive force of liquid metal $F_C$. As the oxide area and adhesion decreases over time after $V_G$ is turned off, the cohesion of liquid metal overcomes the resistance after the time period $\Delta t_{ox}$. Subsequently, the liquid metal filament begins to retract at a rate of $c_1$, which is approximately $0.81 \pm 0.09$ mm·s$^{-1}$. The results indicate that this retracting rate is ~28% smaller than the extending rate $e_1$ (-1.13 mm·s$^{-1}$) (Fig. 4a(ii)), thereby further reinforcing the memory property to some extent.

Benefiting from its memory property, the LLM can perform counting operations of the short-time triggers similar to flytraps, as demonstrated in Fig. 4b and Supplementary Movie 2. Briefly, when the second trigger occurs within a time period $\Delta t \leq \Delta t_{ox}$ (case i), the liquid metal filament elongates and connects the cathode without undergoing any withdrawal. On the other hand, when $\Delta t_{ox} < \Delta t \leq \Delta t_r$ (case ii), the liquid metal filament retracts for a short time period ($\Delta t - \Delta t_{ox}$) at a rate $c_1$, and then elongates at a rate $e_2$ to connect the cathode. Here, $\Delta t_r$ represents the critical value at which the liquid metal barely touches the cathode after the second trigger. It can be calculated

as $3.73 \pm 0.14$ s by evaluating $\int_0^{\Delta t_{ON}} e_1(t)dt - \int_{\Delta t_{ON} + \Delta t_{ox}}^{\Delta t_{ON} + \Delta t} c_1(t)dt + \int_{\Delta t_{ON} + \Delta t}^{2\Delta t_{ON} + \Delta t} e_2(t)dt$ according to Eq. (2).

Figure 4c presents the logic diagram that illustrates the operation of the LLM in distinguishing, memorizing, and counting triggers and generating the corresponding control signals. Initially, the LLM validates the trigger based on the voltage value and proceeds to classify it as either a long-time or short-time trigger based on its duration. If the input is a long-time trigger, the LLM enters a conducting state and outputs the active control signal. On the other hand, if the input is a short-time trigger, the LLM transitions into a standby state, awaiting another trigger. In this scenario, only if the subsequent trigger occurs within the time period $\Delta t_r$, it is considered as a validated second trigger, prompting the LLM to enter the conducting stage and generate the active control signal.

## LLM-based artificial flytrap and potential logic controller
To showcase the capabilities of the LLM, we devise an artificial flytrap system, as depicted in Fig. 5a, this system comprises a voltage supply,

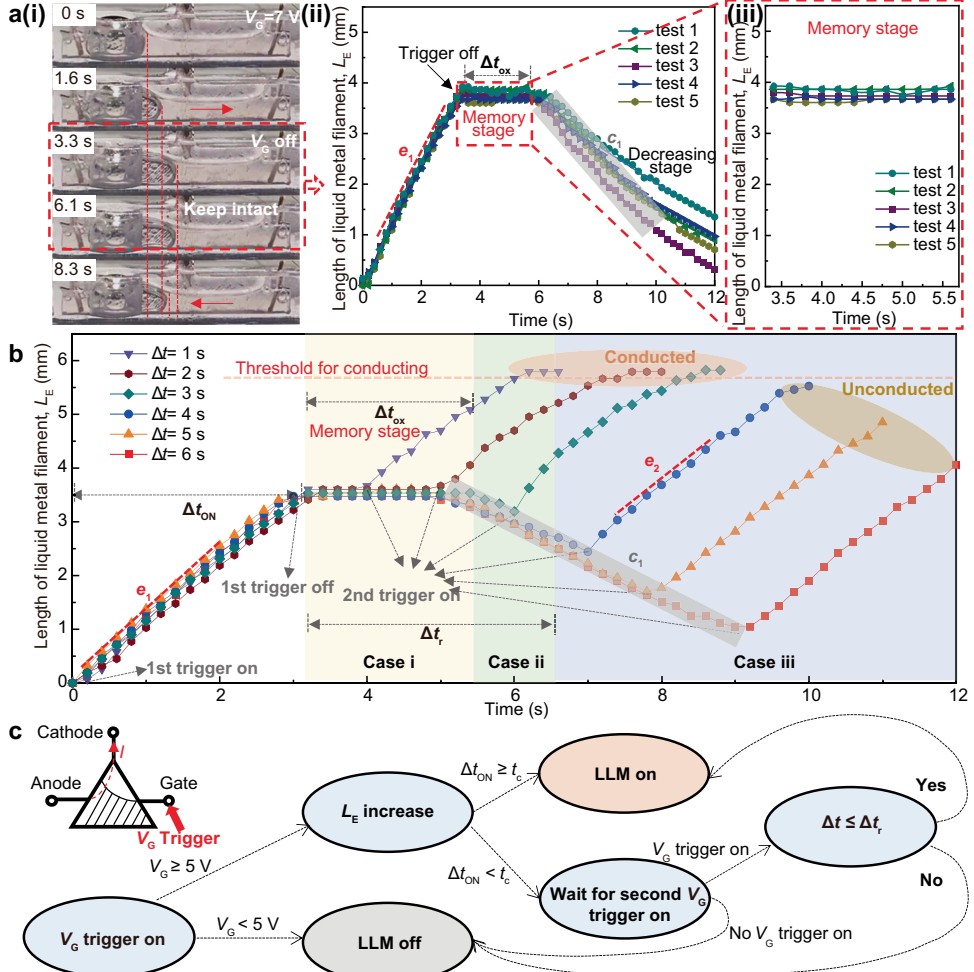

**Fig. 4 | The memory and counting properties of LLM for logic control. a** Defining the duration of the short-time stimulus (gate voltage $V_G$ = 7 V) as 3 s, the liquid metal filament will increase during it (variation rate $e_1$) and be withdrawn with decreasing length (variation rate $c_1$) after that. However, intriguingly, the liquid metal filament remains in place for a certain period of time $\Delta t_{ox}$ about 2.5 s (red dashed box), rather than retracting immediately. **b** When the second trigger occurs within a time period $\Delta t \leq \Delta t_{ox}$ (case i), the liquid metal filament elongates and connects the cathode without undergoing any withdrawal. On the other hand, when $\Delta t_{ox} < \Delta t \leq \Delta t_r$ as case ii, the liquid metal filament retracts for a short time period ($\Delta t$ - $\Delta t_{ox}$) and then elongates to connect the cathode. In case iii that $\Delta t > \Delta t_r$, the LLM could not be conducted. $\Delta t_r$: the threshold of the time interval between two short-time triggers for conducting. **c** The logic diagram of the LLM in distinguishing, memorizing, and counting triggers and generating the corresponding control signals. $I$: current output; $\Delta t_{ON}$: the time duration of the electric trigger.

an electric switch-based artificial sensory hair, and a soft electric actuator-based artificial petal, all of which are connected to the anode, gate, and cathode of the LLM, respectively. Remarkably, the LLM demonstrates the ability to process triggers from the sensory hair and effectively control the bending state of the artificial petal without the need for additional computation or memory units. Further information regarding the design and mechanisms of the electric switch and artificial petal can be found in the supplementary information, specifically in Supplementary Figs. 9–12.

The experimental results confirm the ability of the artificial system to emulate the closure process of the flytrap in response to various triggers. As depicted in Supplementary Fig. 13, when a mechanical trigger of long duration ($t$ = 6 s) is applied to the electric switch, the $V_G$ of the LLM undergoes a change to 7 V, resulting in the extension and connection of the liquid metal to the cathode. Consequently, the output signal manifests as current flowing from the anode to the cathode, thereby actuating the artificial petal to bend and close. In addition, when two individual short-duration mechanical triggers, separated by a 1 s gap, are applied, the artificial petal remains open until the LLM receives the second trigger and initiates conduction (Fig. 5b, c, and Supplementary Movie 3). Notably, the recorded

acceleration of the extending liquid metal, as shown in Fig. 5d, exhibits a resemblance to the action potential observed in biological signals. It exhibits a sharp upward spike followed by a rapid decline, mirroring the characteristic waveform of action potentials in biological systems[20]. These features highlight the potential of the LLM for the development of bioelectronics.

Furthermore, the LLM, with its flytrap-like intelligence, can serve as a functional equivalent to an integrated memristor/transistor. Similar to a memristor, the electronic properties of the LLM are coupled in the presence of an applied electric field. However, beyond a conventional memristor, which is a passive element requiring integration with active circuit elements such as transistors to enable logic functionality, the LLM combines the characteristics of both memristors and transistors within a single device. This hybrid functionality, encompassing the attributes of memristors and transistors, can be conveniently achieved using a single LLM, as depicted in Fig. 5e and Supplementary Movie 4. The equivalent circuit comprises two transistors, one memristor, one resistor, and one capacitor. Specifically, the first and second transistors are corresponding to the evaluation of $V_G$ to LLM for the extension of liquid metal filament and the length of filament for connecting the cathode. The resistance

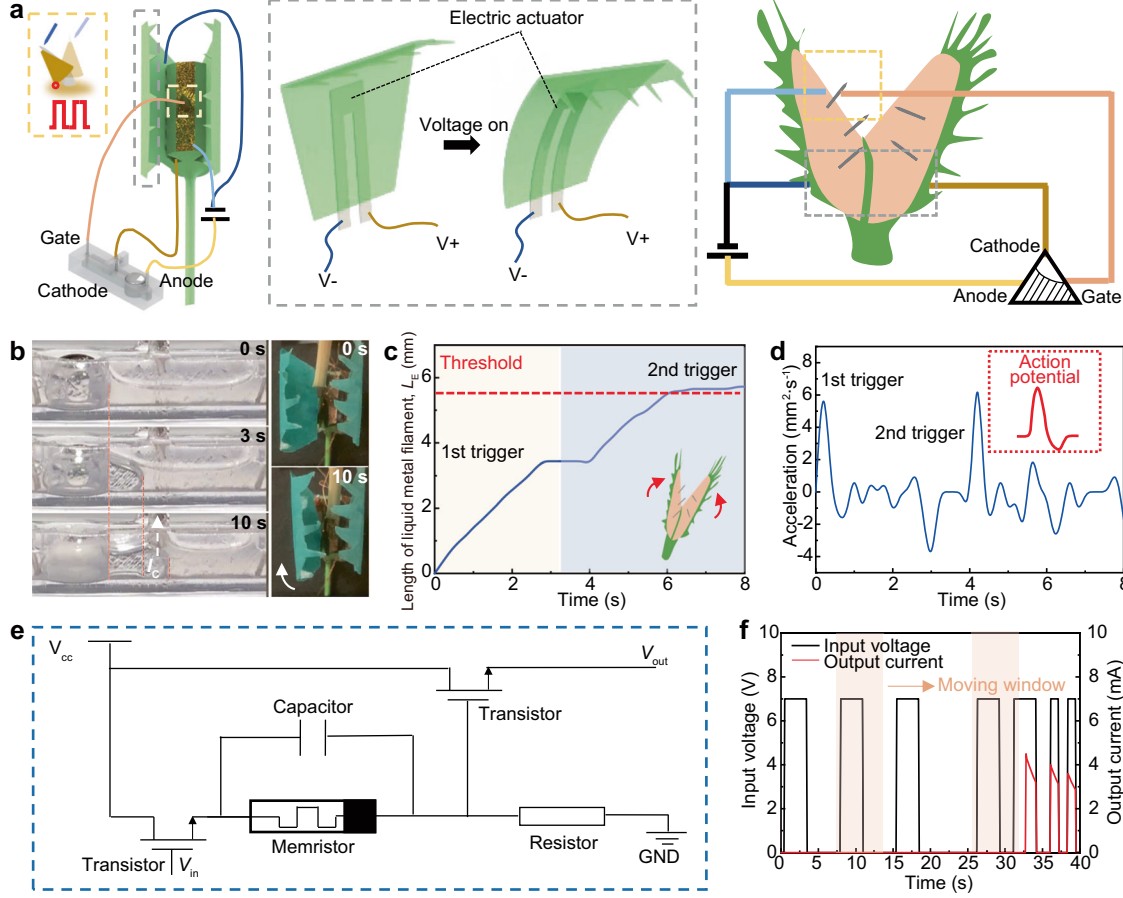

**Fig. 5 | LLM-controlled artificial flytrap and LLM's potential applications. a** The LLM-controlled artificial flytrap is developed for demonstration, which is composed of electric switch-based artificial sensory hair and soft electric actuator-based artificial petals. V+ and V-: applied voltage on artificial petal. **b, c** When two individual short-duration mechanical triggers, separated by a 1 s gap, are applied, the artificial petal remains open until the LLM receives the second trigger and initiates conduction for petal close. **d** The recorded acceleration of the extending liquid metal exhibits a resemblance to the action potential observed in biological signals. **e** The equivalent circuit comprises two transistors, one memristor, one resistor, and one capacitor. **f** The LLM can be integrated as an integrated high-pass filter by utilizing moving window statistics. $V_{CC}$ positive supply voltage; $V_{in}$ input voltage, $V_{out}$ output voltage, GND ground.

variation of the memristor is equivalent to the length change of liquid metal filament. When receiving a pulse voltage $V_{in}$ to the gate, the transistors and memristor would estimate the trigger modes and allow a long-time trigger for conducting. For a short-time trigger that $V_{in}$ is terminated before the second transistor conducts, the capacitor resets the memristor gradually as the liquid metal filament in LLM withdraws, awaiting the next applied pulse. Akin to LLM, only if the second pulse is within the discharging period, it is considered valid.

The simplification achieved by the LLM holds immense potential for a wide range of applications (Supplementary Movie 4). One notable application is its use as an integrated high-pass filter by utilizing moving window statistics. It will accumulate the values of a signal within a moving time window to compare with a predefined threshold, which could be used in applications that involve noise reduction, event detection, energy measurement, and more. This is attributed to the fact that the LLM can only be conducted when the received signal's time period is sufficiently within the moving time window. As illustrated in Fig. 5g, the LLM could filter out signals with long time intervals, as they are unable to turn on the LLM. Furthermore, given its memristor/transistor hybrid nature, the LLM holds potential for applications in the field of neuromorphic systems, specifically serving as a synapse. For instance, it could be integrated into crossbar architectures as the synapse bridge in the layers of neural networks (e.g., the convolutional layer and the pooling layer),

storing and adjusting the weight value (Supplementary Fig. 14). Additionally, the LLM exhibits promise for future use in robotic control and other related domains, allowing for efficient computing independently by replacing the complex integrated electronic components and circuits.

## Discussion

In this study, we introduce a liquid metal-based module inspired by the Venus flytrap, offering an approach for emulating the logical behavior of advanced plants. The LLM leverages the shape-changeable nature of a liquid metal filament, showcasing intriguing memory capabilities. Through the manipulation of the electric field, the length of the liquid metal filament can be controlled based on the electrochemical effect, influenced by both the voltage trigger and its prior shape. With the LLM, we can demonstrate the intelligent biomimetic behavior of the flytrap's bio-response, effectively emulating its stimulus-responsed logic processes. This highlights the potential of LLM in the realm of advanced bioelectronics design.

However, it is worth noting that the current LLM device has certain limitations, including a relatively large trigger duration and a larger physical size. Nevertheless, we believe that these challenges can be addressed through further research. Potential solutions may involve reducing the size of the device by using the micromachining fabrication method, chemical modification of liquid metal to change its electric property, modifying the channel's surface to optimize the

liquid flow resistance, adjusting the electrode distance and liquid metal's surface tension[21], or even exploring alternative materials with self-recovery properties. Although there is still a gap between the demonstrated device and a practical product, we believe that the conceptual framework of a flytrap-inspired logic module will inspire ideas for intelligent component design and pave the way for logic devices.

## Methods

### Construction of LLM

The chip device of LLM employed 3D printing technology in its fabrication process. It primarily comprised a mushroom-shaped channel featuring a reservoir linked to a pathway. The reservoir served as a receptacle for 100 μL of EGaIn, while the remaining portion of the channel was filled with a 1 mol/L NaOH solution (1 mm in height). Within this configuration, two Pt electrodes were inserted into the NaOH solution, positioned at the two ends of the channel, functioning as the anode and gate electrodes respectively. Additionally, another Pt electrode was positioned between the anode and gate electrodes, situated 1 mm above the level of the NaOH solution, acting as the cathode electrode of LLM.

### Fabrication of electric actuator-based artificial flytrap petal

The silver nanowires (Ag NWs) - platinum (Pt) - paper/ polydimethylsiloxane (PDMS) bilayer soft actuator was fabricated acting as the artificial petal. First, the paper sheet was coated with 100 nm Pt layer using the sputtering system (QUORUM #Q150TS Dual target sputtering system). Ag NWs (length 20–50 μm and diameter 115 nm) with 0.5 wt % dispersion in isopropyl alcohol (IPA) were sonicated for 30 min and subsequently centrifuged at 3000 x $g$ for 5 min. The IPA was then replaced by ethylene glycol (EG), and the Ag NWs were redispersed in EG by stirring the solution for 5 min. After that, the Ag NW solution was drop-cast on the Pt-coated paper sheet and dried under light heating. Then, the PDMS was prepared at a 10:1 weight ratio (base: curing agent) and under spinning coating on the opposite side of the obtained paper at the speed of 500 rpm for 15 s and 900 rpm for 30 s. The PDMS was cured at the temperature of 75 °C for 2 h to obtain the Ag NWs-Pt-paper/PDMS composite film. The artificial flower petal can then be fabricated by cutting with a specific shape.

### Driven and characterization of the electric actuator and liquid metal

The voltage was supplied by a DC power supply (Agilent E3611A, 0–20 V). Optical images and videos of liquid metal were captured by the digital microscope (Dino-Lite Digital Microscope). To collect the current signal, we connected a resistor (1 kΩ) to the cathode of the liquid system and collected its voltage using an oscilloscope (Tektronix TBS 1104) as a current reference.

## Data availability

All data that support the findings of this study are available within this article and its Supplementary Information.

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

## Acknowledgements

This work was supported by Hong Kong RGC General Research Fund 11216421, National Natural Science Foundation of China (NSFC)/ Research Grants Council (RGC) Fund N_HKUST638/23, and Shenzhen-Hong Kong-Macau Science and Technology Project (Category C) SGDX20201103093003017. Y.Y. acknowledges the support by the Xiaomi Young Talents Program.

## Author contributions

Y. Y. did the experiment and analyzed the data. Y. S. initialized the idea, designed the experiment, and analyzed the data. All the authors wrote, read, and proofread the manuscript.

## Competing interests

The authors declare no competing interests.
