## [Peer Review File · Nature Communications]

REVIEWER COMMENTS

Reviewer #1 (Remarks to the Author):

The article by Yang and Shen investigated a logic module based on the deformation of liquid metal under electric field. The module provided a wonderful simulation of flytrap behavior, including two quick touches and a single long touch. This work had solid implications for the design of smart logic components and should be available to the literature. Before I suggest this article for publication, I would like the authors to address the following comments.

1. There have been previous studies closely related to liquid metal logic devices which are not mentioned in this manuscript (J. Wissman. *Adv Sci.* 2017, 4, 1700169; D. Li. *Adv Intell Syst.* 2021, 3, 2000246; Y. Wang. *Adv Func. Materials.* 2023, 33, 2303312). It is suggested to compare this work with previous works and discuss its innovations.
2. Equation 2 is the counterpart of Equation 1. Could the formulas corresponding to the symbols e_1 , c_1 , e_2 be more specific, including the relevant factors and mathematical relationships? Besides, the liquid metal is in a channel filled with NaOH solution, its interaction with the surrounding solution or channels has to be taken into account as well.
3. On page 3, according to Lippmann equation, the relationship between the surface tension γ and the potential difference V on localised EDL is $\gamma = \gamma_0 - 1/2 CV^2$, which needs to be added. And Equation 4 is based on the Young-Laplace equation, which is not clearly stated in the manuscript. Small diameter liquid metals tend to move towards the anode under an electric field due to the surface tension gradient. On the other hand, large volume liquid metal tends to move towards the cathode, it differs from the small volume in that it involves forces induced by chemical reactions and this part of the explanation is not clear enough. Besides, whether the viscous resistance of the surrounding fluid is taken into account in the extension and retraction of the liquid metal? It would be better to add diagrams analysing the individual forces on the liquid metal. For more details, the authors can refer to the article (M. Wang. *Phys Chem Chem Phys.* 2017, 19, 18505-18513; X. Yang. *Sci China Technol Sci.* 2016, 59, 597-603).
4. Extending speed and width of channel are negatively correlated in Figure S4. This result is not consistent with the "PLM is positively correlated to w " mentioned on page 3.
5. The manuscript does not indicate what concentration of NaOH was used in the experiment. The concentration of NaOH affects the production of the oxide layer.

6. It is somewhat hard to understand Fig. S12, can the authors explain it in detail?

7. Please check the statement on page 5, penultimate line 4.

Reviewer #2 (Remarks to the Author):

This is an interesting method to emulate the physical system of the Venus flytrap. The authors have made interesting material choices to create a system with some inherit memory and transistor-like functionality (the liquid-metal logic module, LLM). Supporting experiments, data, and analysis are included.

I believe this paper would be significantly strengthened if the following comments are addressed:

1. It seems like the anode is not directly in contact with the liquid metal. This also seems to be true when the gate is triggered, and the liquid metal directly contacts the cathode. Is this correct?
2. If #1 is correct, why isn't the anode in direct contact with the liquid metal? Please elaborate on this point.
3. If #1 is correct, and the anode is not in direct contact with the liquid metal, then the current measured (IC) must also flow through the NaOH surrounding the liquid metal. How do the properties of the NaOH (molarity, electrical conductivity) affect the performance of the LLM?
4. Why/how were the particular dimensions of the LLM's fluidic channels chosen? How important are the dimensions of the cylindrical chamber? How do the dimensions of the chamber affect the LLM performance?
5. Is the fluidic channel "open" (no ceiling) or enclosed (ceiling, sidewalls, floor/substrate)? If the channel is enclosed, where does the NaOH go when it is displaced by the moving liquid metal?
6. As the authors explicitly mention in the Discussion section, the time scale scale of triggering of the LLM is relatively long (on the order of several seconds). This may be too long for some of the proposed applications. What would be the theoretical minimum triggering time?
7. This paper is using electrocapillarity to change the shape of the liquid metal. Thus, this paper should be cited: Khan et al., "Giant and switchable surface activity of liquid metal via surface oxidation," PNAS, 2014, <https://doi.org/10.1073/pnas.1412227111>.

8. Please quantify the number of measurements used to create the graph in Fig. 3e. (There are error bars, so there should be multiple measurements.)
9. The caption of Fig. S4 is incorrect, as it contradicts equation 7 and the observable trend of the data.
10. It seems like there is a missing citation in the Discussion section, after “ionic materials”.
11. Minor point: there is a typo on page 2: “The module comprises three nodes (anode, gate, anode)” should be “anode, gate, cathode”.
12. Minor point: there is a typo in Figure 2a: “Drain” is misspelled.

Reviewer #3 (Remarks to the Author):

The authors proposed a module based on liquid metals to emulating the intelligent preying logic of flytrap in this manuscript. They claimed 'flytrap's "intelligence" arise purely from the proper organization of simple materials and structure, in stark contrast to the complex nervous systems of animals.'. But based on the descriptions in this manuscript, the flytrap's intelligence is quite similar to the synaptic behavior of animals (e.g., paired-pulse facilitation). “The LLM itself exhibits memory and counting properties without involving any other electronic components”, which is actually a property of memristors with the capability of in-memory computing, including liquid metal-based memristors. The whole manuscript is poorly organized and related mechanisms are not clearly addressed. Though the concept of liquid-metal-based memristors for controlling is interesting, I cannot recommend it for publication at this stage due to the above-mentioned reasons. The followings are my other concerns:

1. The manuscript is poorly organized and described with a logic mess. For instances, Figure 1 for the introduction part contains too much information, however, related explanations are too simple. There are some duplicated figure panels in Fig. 1 and 2. The logic of Fig. S7 and S8 is in a mess. Explanations for Fig. S7-S10 are too simple, especially for Fig. S10.
2. The authors should explain more for all the figures in the main text which are difficult to follow due to some missing information. For instances, what do the red lines and numbers stand for? It is difficult to imagine the 3D structure of the LLM due to fuzzy schematics and videos. It is better to provide additional figures, such as a cross-sectional schematic.

3. What is $[Ca^{2+}]_{cyt}$ in Line 72? Abbreviations or symbols should be defined where they first appear. What is called "accumulation"? where does it occur? which side of the membrane? Why did the authors define the term of "accumulation/attenuation model"?
4. For metals, they are equipotential body which is a common sense in physics. What does it mean of "the electric potential gradient along the liquid metal VE" (line 119)? Is liquid metal an exception?
5. Related mechanisms are still obscure. How to electrochemically control the capillarity effect-induced length variation? Does the pristine Ga_2O_3 oxide skin of the liquid metals play a role? Negative role or positive role? Any data can support the reaction (line 122)? How to exclude the possibility of electrostatic force-induced extending of the liquid metal?
6. Fig S4 is correct?

RESPONSES TO REVIEWER COMMENTS

Reviewer's comment (Italic Black) and our answer (Normal Blue):

To Reviewer #1: *The article by Yang and Shen investigated a logic module based on the deformation of liquid metal under electric field. The module provided a wonderful simulation of flytrap behavior, including two quick touches and a single long touch. This work had solid implications for the design of smart logic components and should be available to the literature. Before I suggest this article for publication, I would like the authors to address the following comments.*

1. *There have been previous studies closely related to liquid metal logic devices which are not mentioned in this manuscript (J. Wissman. Adv Sci. 2017, 4, 1700169; D. Li. Adv Intell Syst. 2021, 3, 2000246; Y. Wang. Adv Func. Materials. 2023, 33, 2303312). It is suggested to compare this work with previous works and discuss its innovations.*

Response: Thanks for the reviewer's kind suggestion. We have cited these important studies as reference 13-15 and added the discussion about the comparison between our work and them.

Please see **reference 13, 14, 15.**

Please see **page 2:** “Liquid metal, as a kind of eutectic alloy at room temperature, exhibits great potentials as the materials to develop soft electronic components owing to its unique properties of desirable flexibility, high electrical conductivity, and low toxicity. Compared with the conventional conductive material, the liquid metal can change its shape and moves when exposed to external triggers (e.g., electrical, chemical, etc.). Due to these characters, several intriguing liquid metal-based electric devices have been proposed, such as liquid-metal based-electrical switch and logic components¹³⁻¹⁵. These functions are mainly based on the motions of liquid metal, i.e., coalesce, separation, and migration, which provide changeable states as electrical “0” and “1”. Nevertheless, the two-state feature could not meet the requirement for emulating the flytrap-liked logic since the bio-signals are changing dynamically. Thus, the time-related shape deformation of liquid metal should be taken into account. It could be regulated by the external triggers, which is quite similar to the trigger-controlled variation of bio-signals.”

2. *Equation 2 is the counterpart of Equation 1. Could the formulas corresponding to the symbols $e1$, $c1$, $e2$ be more specific, including the relevant factors and mathematical relationships? Besides, the liquid metal is in a channel filled with NaOH solution, its interaction with the surrounding solution or channels has to be taken into account as well.*

Response: Thanks for the reviewer's kind suggestion. We have updated the force analysis by adding the consideration of liquid metal's interaction with surrounding solution and channels. The force analysis diagram of extending liquid metal is shown as Figure R1, which mainly include the electrochemical-induced surface tension force F_{EC} , the viscous force between the droplet and its surrounding electrolyte F_V , the friction between the liquid metal and the substrate f_s , and the friction between the liquid metal and the channel wall f_w . Therefore, during the extending process of liquid metal, the relevant factors corresponding to the e_1 and e_2 of equation 2 could be specified as F_{EC} , F_V , f_s , and f_w . For the withdrawing process, the relevant factors corresponding to the c_1 of equation 2 could be specified as f_s , f_w , viscous force from NaOH F_V , and the cohesive force of liquid metal F_C , as shown in Figure R2. We have updated the force model in the revised manuscript and evaluated the corresponding e_1 , c_1 , e_2 by experiments. Nevertheless, it is hard to modelling the electrochemical-induce surface tension variation and its corresponding F_{EC} . In the future work, we would try to build an exact mathematical model for them by analyzing the electrochemical process in deep. We have added the related discussion in the revised manuscript.

Figure R1. Force analysis diagrams of extending liquid metal. F_{EC} is the driving force from the electrochemical effect-induced surface tension difference, F_V is the viscous force from surrounding electrolyte, f_s is the friction between the liquid metal and the substrate, and f_w is the friction between the liquid metal and the channel wall.

Figure R2. The adhesion of oxide skin to channel-induced remaining state of liquid metal filament after the gate voltage is removed. F_C is the cohesive force of liquid metal, F_V is the viscous force from surrounding electrolyte, f_s is the friction between the liquid metal and the substrate, and f_w is the friction between the liquid metal and the channel wall. When V_G is turned off, the adhesion of the oxide film to channel will lead to a large friction force f_w and f_s that the liquid metal filament keeps intact. As the adhesion decreases over time, F_C would overcome the resistance leading to the withdraw of liquid metal filament.

Please see Supplementary Fig. 1 and Supplementary Fig. 8.

Please see page 3: “The force analysis diagram of extending liquid metal is shown in Supplementary Fig. 1, which mainly include the electrochemical-induced surface tension force F_{EC} , the viscous force between the droplet and its surrounding electrolyte F_V , the friction between the liquid metal and the substrate f_s , and the friction between the liquid metal and the channel wall f_w . Therefore, during the extending process of liquid metal, the relevant factors corresponding to the e_1 and e_2 of equation (2) could be specified as F_{EC} , F_V , f_s , and f_w .”

Please see page 4: “The relevant factors corresponding to the c_1 of equation (2) could be specified as f_s , f_w , viscous force from NaOH F_V , and the cohesive force of liquid metal F_C .”

Please see page 5: “Subsequently, the liquid metal filament begins to retract at a rate of c_1 , which is approximately $0.81 \pm 0.09 \text{ mm}\cdot\text{s}^{-1}$. The results indicate that this retracting rate is $\sim 28\%$ smaller than the extending rate e_1 ($\sim 1.13 \text{ mm}\cdot\text{s}^{-1}$) (Fig. 4a(ii)), thereby further reinforcing the memory property to some extent.

Please see page 5: “On the other hand, when $\Delta t_{ox} < \Delta t \leq \Delta t_r$ (case ii), the liquid metal filament retracts for a short time period ($\Delta t - \Delta t_{ox}$) at a rate c_1 , and then elongates at a rate e_2 to connect the cathode. Here, Δt_r represents the critical value at which the liquid metal barely touches the

cathode after the second trigger. It can be calculated as 3.73 ± 0.14 s by evaluating

$$\int_0^{\Delta t_{ON}} e_1(t) dt - \int_{\Delta t_{ON} + \Delta t_{OX}}^{\Delta t_{ON} + \Delta t} c_1(t) dt + \int_{\Delta t_{ON} + \Delta t}^{2\Delta t_{ON} + \Delta t} e_2(t) dt$$

according to equation (2).”

3. On page 3, according to Lippmann equation, the relationship between the surface tension γ and the potential difference V on localised EDL is $\gamma = \gamma_0 - 1/2 CV^2$, which needs to be added. And Equation 4 is based on the Young-Laplace equation, which is not clearly stated in the manuscript. Small diameter liquid metals tend to move towards the anode under an electric field due to the surface tension gradient. On the other hand, large volume liquid metal tends to move towards the cathode, it differs from the small volume in that it involves forces induced by chemical reactions and this part of the explanation is not clear enough. Besides, whether the viscous resistance of the surrounding fluid is taken into account in the extension and retraction of the liquid metal? It would be better to add diagrams analysing the individual forces on the liquid metal. For more details, the authors can refer to the article (M. Wang. *Phys Chem Chem Phys*. 2017, 19, 18505-18513; X. Yang. *Sci China Technol Sci*. 2016, 59, 597-603)

Response: Thanks for the reviewer’s kind suggestion. We have revised the manuscript as following:

- We have reorganized the mathematic analysis in the revised manuscript, including adding the Lippmann equation and Young-Laplace equation in the revised manuscript.
- Thanks for the review’s kind reminder, we have added the comment of volume-related extending direction in the revised manuscript by referring to these two valuable references. Guided by them, we added the diagrams as Figure R3 to analyze the individual forces on the liquid metal, including the viscous resistance of the NaOH solution.

Figure R3. Force analysis diagrams of extending liquid metal. F_{EC} is the driving force from the electrochemical effect-induced surface tension difference, F_V is the viscous force from surrounding

electrolyte, f_s is the friction between the liquid metal and the substrate, and f_w is the friction between the liquid metal and the channel wall.

Please see **reference 17,18**.

Please see **Supplementary Fig. 1** and **Supplementary Fig. 8**.

Please see **page 3 and 4**: The relationship between the interface tension and the voltage difference across the EDL is represented by Lippmann's equation, that

$$\gamma = \gamma_0 - \frac{C}{2}V^2 \quad (3)$$

Where γ is the surface tension of the liquid metal, γ_0 is the intrinsic surface tension at the potential of zero charge, C is the per-unit capacitance across the EDL, and V is the electrical potential across liquid metal–electrolyte interface. The pressure difference across the EDL can be obtained from the Young–Laplace's equation:

$$P = \gamma \left(\frac{1}{r_1} + \frac{1}{r_2} \right) \quad (4)$$

where P is the pressure difference between the electrolyte and the liquid metal, r_1 and r_2 are the radius of liquid metal. According to equation (3) and (4), the surface tension should increase in the direction from anode to gate that the pressure on right part of liquid metal would be larger than left part, leading to an extending pressure P_{LM} to the anode direction. Nevertheless, the existence of electrochemical effect would change the direction of surface tension gradient, especially for the liquid metal with large volume^{17,18}. Due to the voltage applied between anode and gate, the entrance of the pathway would attract a large amount of OH^- and promoting an oxidation reaction that leads to the formation of an oxide film on the liquid metal (Fig. 3a (i)):

As the Ga_2O_3 oxide film is hydrophilic that the surface tension would be highly reduced. The liquid metal would tend to increase its contact area with the surrounding electrolyte, resulting in the generation of extending pressure P_{LM} to the gate direction that facilitates elongation of the liquid metal droplet within the channel. When $V_G < 5$ V, surface tension-induced pressure P_{LM} is not sufficient to overcome the capillary resistance P_C (Fig. 3a (ii)), preventing the liquid metal from extending into the pathway. On the other hand, when $V_G \geq 5$ V, P_{LM} becomes strong enough to exceed P_C , enabling the liquid metal to gradually extend into the pathway, as shown in Fig. 3a (iii) and the red dashed line (form A) in Fig. 3b. The force analysis diagram of extending liquid metal is shown in Supplementary Fig. 1, which mainly include the electrochemical-induced surface tension force F_{FC} , the viscous force between the droplet and its surrounding electrolyte F_V ,

the friction between the liquid metal and the substrate f_s , and the friction between the liquid metal and the channel wall f_w . Therefore, during the extending process of liquid metal, the relevant factors corresponding to the e_1 and e_2 of equation (2) could be specified as F_{EC} , F_V , f_s , and f_w .”

4. *Extending speed and width of channel are negatively correlated in Figure S4. This result is not consistent with the "PLM is positively correlated to w" mentioned on page 3.*

Response: Thanks for the reviewer’s kind comment. We are very sorry for the typos in the statement of that. According to experiment result, the extending speed is negatively correlated to the width of channel. Before extending to different channels, the liquid metal stays in the reservoir, of which the pressure and potential flow could be assumed as the same. Suppose droplets of the same mass flow through different channels with cross-sectional areas S_1 and S_2 at velocities v_1 and v_2 respectively, S_1v_1 should be equal to S_2v_2 . When $S_1 < S_2$, $v_1 > v_2$. Therefore, the extending speed of liquid metal filament is smaller when liquid metal is placed in wider channel. We have revised the statement and the caption of this figure.

Please see the **caption of Supplementary Fig. 4** in the revised manuscript: “**Supplementary Figure 4.** The extending speed of liquid metal filament within different channels. Before extending to different channels, the liquid metal stays in the reservoir, of which the pressure and potential flow could be assumed as the same. Suppose droplets of the same mass flow through different channels with cross-sectional areas S_1 and S_2 at velocities v_1 and v_2 respectively, S_1v_1 should be equal to S_2v_2 . When $S_1 < S_2$, $v_1 > v_2$. Therefore, the extending speed of liquid metal filament is smaller when liquid metal is placed in wider channel. Data are presented as mean values \pm SD, the number of independent experiments $n = 5$.”

5. *The manuscript does not indicate what concentration of NaOH was used in the experiment. The concentration of NaOH affects the production of the oxide layer.*

Response: Thanks for the reviewer’s kind comment. The concentration of NaOH used is 1 mol/L. We have added it in the revised manuscript.

Please see **page 3**: “100 μ L Eutectic gallium–indium (EGaIn) is placed in the reservoir and the rest of the cavity is filled with 1 mol/L NaOH solution (1 mm in height).”

6. *It is somewhat hard to understand Fig. S12, can the authors explain it in detail?*

Response: Thanks for the reviewer’s kind comment. Fig. S12 is the schematic of the LLM’s potential application in neural networks. The LLMs could be placed at the junctions of the

crossbar to carry out logic operations. The outputs of the crossbar depend on the applied voltages to the rows and the columns of the crossbar. Due to the integrated memristor and transistor function of LLM, storage and computation can both be carried out. The crossbar nodes' conducting state could then be used as the weights of neural network for training. We have added the related statement in the revised manuscript.

Please see the **caption of Supplementary Fig. 14** (former Fig. S12): **“Supplementary Figure 14. The schematic of the potential neuromorphic applications of LLM. The LLMs could be placed at the junctions of crossbar to carry out logic operations. The outputs of the crossbar depend on the applied voltages to the rows and the columns of the crossbar. Due to the integrated memristor and transistor function of LLM, storage and computation can both be carried out. The crossbar nodes' conducting state could then be used as the weights of neural network for training.”**

7. *Please check the statement on page 5, penultimate line 4.*

Response: Thanks for the reviewer's kind comment. We have revised the mistake.

Please see **page 6**: **“Potential solutions may involve reducing the size of device by using the MEMS fabrication method, chemical modification of liquid metal to change its electric property, modifying the channel's surface to optimize the liquid flow resistance, adjusting the electrode distance and liquid metal's surface tension²⁰, or even exploring alternative materials with self-recovery properties.”**

Reviewer #2: *This is an interesting method to emulate the physical system of the Venus flytrap. The authors have made interesting material choices to create a system with some inherit memory and transistor-like functionality (the liquid-metal logic module, LLM). Supporting experiments, data, and analysis are included. I believe this paper would be significantly strengthened if the following comments are addressed:*

1. *It seems like the anode is not directly in contact with the liquid metal. This also seems to be true when the gate is triggered, and the liquid metal directly contacts the cathode. Is this correct?*

Response: Thanks for the reviewer's kind comment. Yes, you are right. The anode is not directly in contact with the liquid metal all the time. As shown in Figure R4, the liquid metal would contact the anode in the first few seconds when voltage is applied to the gate and anode. During the extending process of liquid metal, it separates with the anode gradually. When the cathode is conducted, the liquid metal is not directly in contact with the anode while directly contacts the cathode. We have added the related content in the revised manuscript.

Figure R4. The contact state between liquid metal and electrodes during the working process.

Please see **Supplementary Fig. 5**.

Please see **page 4**: “The liquid metal is in contact with the anode at the first few seconds when voltage is applied on gate and anode (Supplementary Fig. 5). Since the volume of liquid metal is constant, during the extending process of liquid metal filament, the left end of liquid metal would

gradually separate with the anode. When the cathode is conducted, the liquid metal is not directly in contact with the anode while directly contacts the cathode.”

2. *If #1 is correct, why isn't the anode in direct contact with the liquid metal? Please elaborate on this point.*

Response: Thanks for the reviewer's kind comment. The liquid metal is in contact with the anode in the first few seconds when voltage is applied to the gate and anode. Since the volume of liquid metal is constant, during the extending process of the liquid metal filament, the left end of the liquid metal would gradually separate with the anode. We have elaborated on it in the revised manuscript.

Please see **Supplementary Fig. 5**.

Please see **page 4**: “The liquid metal is in contact with the anode at the first few seconds when voltage is applied on gate and anode (Supplementary Fig. 5). Since the volume of liquid metal is constant, during the extending process of liquid metal filament, the left end of liquid metal would gradually separate with the anode. When the cathode is conducted, the liquid metal is not directly in contact with the anode while directly contacts the cathode.”

3. *If #1 is correct, and the anode is not in direct contact with the liquid metal, then the current measured (I_C) must also flow through the NaOH surrounding the liquid metal. How do the properties of the NaOH (molarity, electrical conductivity) affect the performance of the LLM?*

Response: Thanks for the reviewer's kind comment. We have conducted experiment to verify how the properties of NaOH affect the performance of the LLM. As the electrical conductivity of a NaOH solution depends on its concentration, the molarity and electrical conductivity of NaOH are positively related. Thus, we evaluate the influence by adjusting the molarity as 0.5-1.5 mol/L to conduct the experiments. The experiment results are shown as Figure R5, that the conducted current would increase with higher molarity of NaOH.

Figure R5. The conducted current with different molarity of NaOH. Since the liquid metal is not direct connected to the anode, the conducted current is positively related to the molarity of NaOH.

Please see **Supplementary Fig. 7**.

Please see **page 4**: “The results show that the current \$I_c\$ stabilizes quickly as the flow of the liquid metal becomes steady after achieving conduction, and its value is positively correlated with the gate voltage \$V_G\$ and the molarity of NaOH, in good agreement with the theoretical prediction (Fig. 3c, Fig. 3d, and Supplementary Fig. 7).”

4. *Why/how were the particular dimensions of the LLM’s fluidic channels chosen? How important are the dimensions of the cylindrical chamber? How do the dimensions of the chamber affect the LLM performance?*

Response: Thanks for the reviewer’s kind comment. The chamber and channel are designed based on the purpose that the liquid metal within the chamber could be full enough to flow to the channel and reach the cathode. Therefore, we designed the chamber with a larger dimension compared with the channel. The exact dimensions of the cylindrical chamber are not that important, which have little influence on the performance of the device. As shown in Figure R6, we test the performance of the device with larger and smaller dimensions of the chamber compared with origin raids $r = 2.2$ mm, that $r_s + 1$ mm = $r = r_l - 1$ mm. The experiment results show that the liquid metal could be manipulated controllably as well. We have added related discussion in the revised manuscript.

Figure R6. The actuation performance of device with dimensions of the cylindrical chamber, including a) $r_s = r - 1 = 1.2$ mm, and b) $r_l = r + 1 = 3.2$ mm.

Please see page 4: “For the consideration that the liquid metal within the reservoir should be full enough to reach the cathode and still reserve certain amount of it within the reservoir to drag the effluent liquid metal back after voltage off, the dimension of reservoir should be larger than the channel width. According to the experiments, the devices perform well over different reservoir dimensions (Supplementary Fig. 6) and in this work $r = 2.2$ mm is used for demonstration.”

Please see Supplementary Fig. 6.

5. *Is the fluidic channel “open” (no ceiling) or enclosed (ceiling, sidewalls, floor/substrate)? If the channel is enclosed, where does the NaOH go when it is displaced by the moving liquid metal?*

Response: Thanks for the reviewer’s kind comment. The fluidic channel is open without a ceiling as shown in Figure R7. The NaOH solution would gather to the right end of channel when it is displaced by the moving liquid metal. We have added the related statement for better demonstration.

Please see page 2: “The LLM mainly consists of an open mushroom-shaped channel with a reservoir and a connected pathway.”

Please see Figure 2a.

Figure R7. Construction of LLM. The LLM mainly consists of a mushroom-shaped channel with a reservoir and a connected pathway. Liquid metal is placed in the reservoir and the rest of the cavity is filled with NaOH solution. Two Pt electrodes are inserted into the NaOH solution as the anode and gate, which are located on the left side and right side of the channel, respectively. Additionally, one Pt electrode between the anode and gate is designed as the cathode.

6. *As the authors explicitly mention in the Discussion section, the time scale of triggering of the LLM is relatively long (on the order of several seconds). This may be too long for some of the proposed applications. What would be the theoretical minimum triggering time?*

Response: Thanks for the reviewer's kind comment. The time for liquid metal to extend to the channel is short that within 1 seconds. Theoretically, if the cathode is very close to the entrance, the triggering time could reach the millisecond level. Nevertheless, in this case the flowing properties of liquid metal such as memory effect would be out of work. Future works could be done to make the liquid metal keep flowing properties in small-scale, including reducing the size of the device by using the MEMS fabrication method, chemical modification of liquid metal to change its electric property and surface tension, etc.

Please see **page 6**: “Potential solutions may involve reducing the size of device by using the MEMS fabrication method, chemical modification of liquid metal to change its electric property, modifying the channel's surface to optimize the liquid flow resistance, adjusting the electrode distance and liquid metal's surface tension²⁰, or even exploring alternative materials with self-recovery properties.”

7. *This paper is using electrocapillarity to change the shape of the liquid metal. Thus, this paper should be cited: Khan et al., “Giant and switchable surface activity of liquid metal via surface oxidation,” PNAS, 2014, <https://doi.org/10.1073/pnas.1412227111>.*

Response: Thanks for the reviewer's kind suggestion. Sorry for missing citing this important work. The proposed electrocapillarity and electrochemical controlling method provides valuable guidance to this work. We have added this reference in the revised manuscript.

Please see **page 3**: "Liquid metal serves as the conductive medium to emulate the potential variation based on the electrochemically controlled capillarity effect-induced length variation¹⁶."

Please see **reference 16**.

8. *Please quantify the number of measurements used to create the graph in Fig. 3e. (There are error bars, so there should be multiple measurements.)*

Response: Thanks for the reviewer's kind suggestion. We have added the number on the revised manuscript, which is 5.

Please see the **caption of Fig. 3e**: "e) The conducting time required for the different gate voltage inputs decreases with higher voltage applied. Data are presented as mean values \pm standard deviation (SD), the number of independent experiments $n = 5$."

9. *The caption of Fig. S4 is incorrect, as it contradicts equation 7 and the observable trend of the data.*

Response: Thanks for the reviewer's kind comment. We are very sorry for the typos in the statement of that. According to experiment result, the extending speed is negatively correlated to the width of channel. Before extending to different channels, the liquid metal stays in the reservoir, of which the pressure and potential flow could be assumed as the same. Suppose droplets of the same mass flow through different channels with cross-sectional areas S_1 and S_2 at velocities v_1 and v_2 respectively, S_1v_1 should be equal to S_2v_2 . When $S_1 < S_2$, $v_1 > v_2$. Therefore, the extending speed of liquid metal filament is smaller when liquid metal is placed in wider channel. We have revised the statement and the caption of this figure.

Please see the **caption of Supplementary Fig. 4** in the revised manuscript: "**Supplementary Figure 4**. The extending speed of liquid metal filament within different channels. Before extending to different channels, the liquid metal stays in the reservoir, of which the pressure and potential flow could be assumed as the same. Suppose droplets of the same mass flow through different channels with cross-sectional areas S_1 and S_2 at velocities v_1 and v_2 respectively, S_1v_1

should be equal to S_2v_2 . When $S_1 < S_2$, $v_1 > v_2$. Therefore, the extending speed of liquid metal filament is smaller when liquid metal is placed in wider channel. Data are presented as mean values \pm SD, the number of independent experiments $n = 5$.

10. *It seems like there is a missing citation in the Discussion section, after “ionic materials”.*

Response: Thanks for the reviewer’s kind comment. Sorry for the typo. We have revised it in the new manuscript.

11. *Minor point: there is a typo on page 2: “The module comprises three nodes (anode, gate, anode)” should be “anode, gate, cathode”.*

Response: Thanks for the reviewer’s kind comment. Sorry for the mistake. We have revised the typo in the manuscript.

Please see **page 2**: “The module comprises three nodes (anode, gate, cathode) with liquid metal in NaOH solution as the conductive medium (Fig. 2a).”

12. *Minor point: there is a typo in Figure 2a: “Drain” is misspelled.*

Response: Thanks for the reviewer’s kind suggestion. We have revised it and check again the typos in the manuscript.

Please see **Figure 1b** (former Figure 2a).

Reviewer #3: *The authors proposed a module based on liquid metals to emulating the intelligent preying logic of flytrap in this manuscript. They claimed 'flytrap's "intelligence" arise purely from the proper organization of simple materials and structure, in stark contrast to the complex nervous systems of animals.'. But based on the descriptions in this manuscript, the flytrap's intelligence is quite similar to the synaptic behavior of animals (e.g., paired-pulse facilitation). "The LLM itself exhibits memory and counting properties without involving any other electronic components", which is actually a property of memristors with the capability of in-memory computing, including liquid metal-based memristors. The whole manuscript is poorly organized and related mechanisms are not clearly addressed. Though the concept of liquid-metal-based memristors for controlling is interesting, I cannot recommend it for publication at this stage due to the above-mentioned reasons. The followings are my other concerns:*

1. *The manuscript is poorly organized and described with a logic mess. For instances, Figure 1 for the introduction part contains too much information, however, related explanations are too simple. There are some duplicated figure panels in Fig. 1 and 2. The logic of Fig. S7 and S8 is in a mess. Explanations for Fig. S7-S10 are too simple, especially for Fig. S10.*

Response: Thanks for the reviewer's kind comment. We have revised the manuscript as following:

- We have recognized the content of Figure 1 and 2 by deleting some duplicated figure panels. Please see Figure R8 and R9.

Figure R8. Venus flytrap-inspired liquid metal-based logic module. a) The flytrap could generate an electric signal after receiving mechanical stimuli to sensory hair, leading to the increase of cytoplasmic calcium ($[Ca^{2+}]_{\text{cyt}}$) concentration. The strength of the electric signal gradually decreased after the first stimulus, then a second stimulus is required to increase the signal to a higher level, meeting the threshold that is correlated to the leaf blade closure. Once the time interval between two stimuli Δt exceeds the threshold t_r , the strength increase induced by the second stimulus will be insufficient to meet the putative threshold for leaf closure. b) The ion concentration could be abstracted out as the signal accumulation/attenuation time-dependent ion diffusion (SAA) model. The SAA model suggests itself be a three-nodes embedded system, the conductive medium of which is required with time-dependent positive and negative potential variation ability corresponding to the gate trigger. Inspired by that, a liquid metal-based logic module (LLM) is proposed. c) The liquid system is designed based on amorphous liquid metal, acting as the controller for the artificial flytrap. Similar to the ion diffusion mechanism within a flytrap, the liquid metal alters its shape based on its surface tension variation. Once the length of liquid metal reaches the threshold, the artificial flytrap petals would receive the corresponding electric signals to implement the closure process.

Figure R9. a) The trigger-dependent closure of artificial petal could be implemented by the LLM. It mainly consists of a mushroom-shaped channel with a reservoir and a connected pathway. Liquid metal is placed in the reservoir and the rest of the cavity is filled with NaOH solution. Two Pt electrodes are inserted into the NaOH solution as the anode and gate, which are located on the left side and right side of the channel, respectively. Additionally, one Pt electrode between the anode and gate is designed as the cathode. b) The ion concentration could be abstracted out as the SAA model. When a fast stimulus is applied on the sensory, the ion level increases (stage i) and then fades off (stage ii) until a second stimulus is applied within the attenuation period to accumulate it to exceed the threshold (stage iii). c) The output of the liquid system is depending on the shape deformation of the liquid metal filament. Two consecutive triggers could lead to the conducting of LLM that the length of liquid metal filament increases (stage i) and decreases (stage ii) when first trigger is applied and withdrawal, and then increases again as the second trigger applied (stage iii).

Please see **Figure 1** and **2**.

- We have changed the order of Fig. S7 and S8 as new Supplementary Fig. 10 and 11.
- We have extended the explanations for Fig. S7-10 on the revised manuscript.

Please see the **caption of Supplementary Fig. 9-12** (former Fig. S7-10): “**Supplementary Figure 9.** Structure and working mechanism of electric switch-based artificial sensory hair. The electric switch is composed of Cu substrate, PDMS “hair”, and a Cu sheet. When the PDMS receives the mechanical trigger, it would bend to make the Cu sheet contact with the Cu substrate. In this case, the electric switch is ON with current flow through.”

“**Supplementary Figure 10.** Fabrication process of electric actuator. First, the paper sheet was coated with Pt layer using the sputtering system. The Ag NW solution was drop-cast on the Pt-coated paper sheet and dried under light heating. Then the PDMS was spinning coated on the opposite side of the obtained paper at the speed of 500 rpm for 15 s and 900 rpm for 30 s. After that, PDMS was cured at the temperature of 75 °C for 2 h to obtain the Ag NWs-Pt-paper/PDMS composite film. The artificial flower petal can then be fabricated by cutting with a specific shape.”

“**Supplementary Figure 11.** Bending mechanism of artificial petal. When applying the DC voltage on the Ag NWs, the generated electrical current is converted into thermal energy via Joule heating, rapidly heating the conductive film. The produced thermal energy is then transferred to paper and PDMS layer. Since PDMS owns high CTE while paper owns a very low CTE value, the bilayer would bend due to the interfacial stress. Once the applied voltage is turned off, the actuator gradually returns to its original shape due to the heat loss.”

“**Supplementary Figure 12.** The bending performance of electric actuator according to voltage applied. Higher voltage would generate higher heat and thermal expansion volume, leading to the higher bending angle (sample size: 5*15 mm). Data are presented as mean values \pm SD, the number of independent experiments $n = 5$.”

2. *The authors should explain more for all the figures in the main text which are difficult to follow due to some missing information. For instances, what do the red lines and numbers stand for? It is difficult to imagine the 3D structure of the LLM due to fuzzy schematics and videos. It is better to provide additional figures, such as a cross-sectional schematic.*

Response: Thanks for the reviewer’s kind suggestion. We have added the related content to make the manuscript more understandable.

- The red line in Figure 2b stands for the signal strength of flytrap and the number stands for three signal variation stage. The number in Figure 2e is corresponding to Figure 2b, showing the flytrap’ signal change-liked length variation of liquid metal. For better demonstration, we have revised Figure 2b and 2e as Figure R10, and added related explanation in the revised manuscript.

Please see **Figure 2** and **page 2**: “Specifically, the first mechanical stimulus to a sensory hair increases $[Ca^{2+}]_{cvt}$ and the elevated $[Ca^{2+}]_{cvt}$ decreases after the first stimulus. The second stimulus additively increases $[Ca^{2+}]_{cvt}$ that meets a putative threshold for movement.”

“As shown in Fig. 2b, when a fast stimulus is applied on the gate, the signal strength (red line, difference between ion flow in and ion efflux) increases (stage i) and then fades off (stage ii) until a second stimulus is applied within the attenuation period to accumulate it to exceed the threshold (stage iii).”

Please see **page 3**: “The LLM relies on the interplay between the amorphous liquid metal filament and the flowing NaOH solution, where the positive and negative length variations of liquid metal filament L_E are regulated by the electric trigger-dependent surface tension variation, the capillary resistance, and the cohesion of liquid metal.”

“Otherwise, two consecutive fast triggers are required that the length of liquid metal filament increases (stage i) and decreases (stage ii) when first trigger is applied and withdrawal, and then increases again as the second trigger applied (stage iii).”

Figure R10. The ion concentration could be abstracted out as the SAA model. When a fast stimulus is applied on the sensory, the signal strength increases (stage i) and then fades off (stage ii) until a second stimulus is applied within the attenuation period to accumulate it to exceed the threshold (stage iii). Similarly, the output of the liquid system is depending on the shape deformation of the liquid metal filament. Two consecutive triggers could lead to the conducting of LLM that the length of liquid metal filament increases (stage i) and decreases (stage ii) when first trigger is applied and withdrawal, and then increases again as the second trigger applied (stage iii).

- To better demonstrate the 3D structure of LLM, we have added the cross-sectional, top-view, and 3D schematics, as shown in Figure R11.

Please see **Figure 2a** and **page 2**: “The module comprises three nodes (anode, gate, cathode) with liquid metal in NaOH solution as the conductive medium (Fig. 2a).”

Please see **page 3**: “The LLM mainly consists of an open mushroom-shaped channel with a reservoir and a connected pathway. 100 μ L Eutectic gallium–indium (EGaIn) is placed in the reservoir and the rest of the cavity is filled with 1 mol/L NaOH solution (1 mm in height). Two Pt electrodes are inserted into the NaOH solution as the anode (source of the liquid system) and gate, which are located on the left side and right side of the channel, respectively. Liquid metal serves as the conductive medium to emulate the potential variation based on the electrochemically controlled capillarity effect-induced length variation¹⁶. The stimulus signal from “sensory hair” is received by the anode and gate electrode in the form of potential difference. Additionally, one Pt electrode between the anode and gate is designed as the cathode (“drain” of liquid system) with a height of 1 mm above the NaOH liquid level, which is used to output signals to the “trap”.”

Figure R11. Construction of LLM. The LLM mainly consists of a mushroom-shaped channel with a reservoir and a connected pathway. Liquid metal is placed in the reservoir and the rest of the cavity is filled with NaOH solution. Two Pt electrodes are inserted into the NaOH solution as the anode and gate, which are located on the left side and right side of the channel, respectively. Additionally, one Pt electrode between the anode and gate is designed as the cathode.

3. *What is $[Ca^{2+}]_{cyt}$ in Line 72? Abbreviations or symbols should be defined where they first appear. What is called "accumulation"? where does it occur? which side of the membrane? Why did the authors define the term of "accumulation/attenuation model"?*

Response: Thanks for the reviewer's kind comment.

- $[Ca^{2+}]_{cyt}$ stands for cytoplasmic calcium. We have added the definition on the text where it first appeared.
- The "accumulation" refers to the gradually increasing process in the amount of $[Ca^{2+}]_{cyt}$. When a sensory hair is mechanical triggered, the accumulation of $[Ca^{2+}]_{cyt}$ occurs at the base of the hair. It is within the membrane. We have added the relation content in the revised manuscript.

Please see **page2**: "When a sensory hair is mechanical triggered, the concentration of cytoplasmic calcium ($[Ca^{2+}]_{cyt}$) within membrane increases at the base of the hair."

- During the triggering process of sensory hair, the concentration of $[Ca^{2+}]_{cyt}$ increases and decreases accordingly, thus we call it "accumulation/attenuation model". Specifically, the first mechanical stimulus to a sensory hair increases $[Ca^{2+}]_{cyt}$ and the elevated $[Ca^{2+}]_{cyt}$ decreases after the first stimulus. The second stimulus additively increases $[Ca^{2+}]_{cyt}$ that meets a putative threshold for movement.

Please see **page2**: "Specifically, the first mechanical stimulus to a sensory hair increases $[Ca^{2+}]_{cyt}$ and the elevated $[Ca^{2+}]_{cyt}$ decreases after the first stimulus. The second stimulus additively increases $[Ca^{2+}]_{cyt}$ that meets a putative threshold for movement."

4. *For metals, they are equipotential body which is a common sense in physics. What does it mean of "the electric potential gradient along the liquid metal VE" (line 119)? Is liquid metal an exception?*

Response: Thanks for the reviewer's kind comment. Liquid metal is equipotential body as well. The "electric potential gradient" in this sentence actually refers to the liquid metal-electrolyte interface along the liquid metal body. Sorry for the misunderstanding. We have revised this sentence for better understanding.

Please see page 3: “The EDL undergoes changes in its charge distribution in response to changes in the electrical potential on the gate V_G , which generates an electric potential gradient V_E along the liquid metal-electrolyte interface.”

5. *Related mechanisms are still obscure. How to electrochemically control the capillarity effect-induced length variation? Does the pristine Ga₂O₃ oxide skin of the liquid metals play a role? Negative role or positive role? Any data can support the reaction (line 122)? How to exclude the possibility of electrostatic force-induced extending of the liquid metal?*

Response: Thanks for the reviewer’s insightful comment. The actuation mechanism of liquid metal is based on the electrochemical effect that the surface oxide plays an essential positive role. We have reorganized the mechanism explanation from these two aspects for better demonstration.

- Liquid metal processes large surface tension that forms a droplet shape within the electrolyte. The way to actuate the liquid metal is to lower its surface tension directionally, which could be made by both electrocapillarity and electrochemical effect. Electrocapillarity lowers the interfacial tension of the metal γ from its maximum value γ_0 due to the potential difference based on the Lippmann’s equation, that $\gamma = \gamma_0 - \frac{C}{2}V^2$, where γ is the surface tension of the liquid metal, γ_0 is the intrinsic surface tension in the absence of the EDL, C is the per-unit capacitance across the EDL, and V is the voltage across liquid metal-electrolyte interface. Therefore, any change in potential will result in a decrease in the surface tension. Normally, electrocapillarity is limited to a modest variation range of interfacial tension, whereas the electrochemical reactions further lower the interfacial tension. When the oxide forms, it replaces this high-energy interface with two new interfaces: metal–metal oxide and metal oxide–electrolyte. Most oxides, including gallium oxide, form hydroxyl groups on their exterior surface, rendering them hydrophilic. Therefore, the surface tension could be dramatically lower for actuation.
- The surface oxide Ga₂O₃ plays a positive role in the mechanism of electrochemical effect. Due to the voltage applied between anode and cathode, the entrance of the pathway would attract a large amount of OH⁻ and promoting an oxidation reaction that leads to the formation of an oxide film on the liquid metal. As the Ga₂O₃ oxide film is hydrophilic that the surface tension would be highly reduced. The liquid metal would

tend to increase its contact area with the surrounding electrolyte, resulting in the generation of extending pressure that facilitates elongation of the liquid metal droplet within the channel.

- The oxide reaction has been widely studied in the actuating of liquid metal, which could be supported by references such as following:
 - Khan, M. R., Eaker, C. B., Bowden, E. F., Dickey, M. D., Giant and switchable surface activity of liquid metal via surface oxidation. *Proceedings of the National Academy of Sciences* **2014**, *111* (39), 14047-14051.
 - Liao, J., Majidi, C. & Sitti, M. Liquid Metal Actuators: A Comparative Analysis of Surface Tension Controlled Actuation. *Advanced Materials* **2023**, 2300560.
 - Liao, J. & Majidi, C. Muscle-Inspired Linear Actuators by Electrochemical Oxidation of Liquid Metal Bridges. *Advanced Science* **2022**, *9*, 2201963.
- The possibility of electrostatic force-induced extending of the liquid metal could be excluded according to the moving direction of liquid metal. When place the metal into the electric field, the electron within metal would tend to move to the opposite direction of potential gradient. Thus, if the electrostatic force works, the liquid metal should be actuated to the anode direction. However, in our experiment, the liquid metal is moving away from the anode. Thus, the electrostatic force could be excluded.

Please see **page 3** and **page 4**: The relationship between the interface tension and the voltage difference across the EDL is represented by Lippmann's equation, that

$$\gamma = \gamma_0 - \frac{C}{2}V^2 \quad (3)$$

Where γ is the surface tension of the liquid metal, γ_0 is the intrinsic surface tension at the potential of zero charge, C is the per-unit capacitance across the EDL, and V is the electrical potential across liquid metal–electrolyte interface. The pressure difference across the EDL can be obtained from the Young–Laplace's equation:

$$P = \gamma \left(\frac{1}{r_1} + \frac{1}{r_2} \right) \quad (4)$$

where P is the pressure difference between the electrolyte and the liquid metal, r_1 and r_2 are the radius of liquid metal. According to equation (3) and (4), the surface tension should increase in the direction from anode to gate that the pressure on right part of liquid metal would be larger than left part, leading to an extending pressure P_{LM} to the anode direction. Nevertheless, the existence of electrochemical effect would change the direction of surface tension gradient, especially for the liquid metal with large volum^{17, 18}. Due to the voltage

applied between anode and gate, the entrance of the pathway would attract a large amount of OH⁻ and promoting an oxidation reaction that leads to the formation of an oxide film on the liquid metal (Fig. 3a (i)):

As the Ga₂O₃ oxide film is hydrophilic that the surface tension would be highly reduced. The liquid metal would tend to increase its contact area with the surrounding electrolyte, resulting in the generation of extending pressure P_{LM} to the gate direction that facilitates elongation of the liquid metal droplet within the channel.

6. *Fig S4 is correct?*

Response: Thanks for the reviewer's kind comment. We are very sorry for the typos in the statement of that. According to experiment result, the extending speed is negatively correlated to the width of channel. Before extending to different channels, the liquid metal stays in the reservoir, of which the pressure and potential flow could be assumed as the same. Suppose droplets of the same mass flow through different channels with cross-sectional areas S_1 and S_2 at velocities v_1 and v_2 respectively, S_1v_1 should be equal to S_2v_2 . When $S_1 < S_2$, $v_1 > v_2$. Therefore, the extending speed of liquid metal filament is smaller when liquid metal is placed in wider channel. We have revised the statement and the caption of this figure.

Please see the **caption of Supplementary Fig. 4** in the revised manuscript: "**Supplementary Figure 4.** The extending speed of liquid metal filament within different channels. Before extending to different channels, the liquid metal stays in the reservoir, of which the pressure and potential flow could be assumed as the same. Suppose droplets of the same mass flow through different channels with cross-sectional areas S_1 and S_2 at velocities v_1 and v_2 respectively, S_1v_1 should be equal to S_2v_2 . When $S_1 < S_2$, $v_1 > v_2$. Therefore, the extending speed of liquid metal filament is smaller when liquid metal is placed in wider channel. Data are presented as mean values \pm SD, the number of independent experiments $n = 5$."

REVIEWERS' COMMENTS

Reviewer #1 (Remarks to the Author):

The authors overall well addressed the review comments and suggestions. I would like to recommend accept the manuscript for publication after finalized minor justification. The core of the present technology is based on electrical manipulation of liquid metal. To be seriously accurate and also for providing complete information for the readers, the following earliest comprehensive findings on diverse electrical manipulation of liquid metal should be cited and discussed in the present work: a) Sheng et al., Diverse transformations of liquid metals between different morphologies, *Advanced Materials*, 2014. All the subsequent electrical actuations of liquid metal are later than that. So this classical literature should not be neglected.

Reviewer #2 (Remarks to the Author):

The authors have done a good job of addressing the comments from the prior round of reviews. I believe this manuscript is much stronger as a result.

Reviewer #3 (Remarks to the Author):

The authors have addressed all the comments properly. Hence I recommend it for publication in *Nature Communications*.

RESPONSES TO REVIEWER COMMENTS

Reviewer's comment (*Italic Black*) and our answer (Normal Blue):

To Reviewer #1: *The authors overall well addressed the review comments and suggestions. I would like to recommend accept the manuscript for publication after finalized minor justification. The core of the present technology is based on electrical manipulation of liquid metal. To be seriously accurate and also for providing complete information for the readers, the following earliest comprehensive findings on diverse electrical manipulation of liquid metal should be cited and discussed in the present work: a) Sheng et al., Diverse transformations of liquid metals between different morphologies, Advanced Materials, 2014. All the subsequent electrical actuations of liquid metal are later than that. So this classical literature should not be neglected.*

Response: Thanks for the reviewer's kind recommendation and reminder. We have cited this important study as reference 13.

Please see **reference 13**.

Please see **page 2**: “Compared with the conventional conductive material, the liquid metal could be manipulated by external triggers (e.g., electrical, chemical, etc.), and then transform between different morphologies or move¹³.”

Reviewer #2: *The authors have done a good job of addressing the comments from the prior round of reviews. I believe this manuscript is much stronger as a result.*

Response: Thanks for the reviewer's kind approval.

Reviewer #3: *The authors have addressed all the comments properly. Hence I recommend it for publication in Nature Communications.*

Response: Thanks for the reviewer's kind recommendation.